# Uptake Rates of Risk-Reducing Surgeries for Women at Increased Risk of Hereditary Breast and Ovarian Cancer Applied to Cost-Effectiveness Analyses: A Scoping Systematic Review

**DOI:** 10.3390/cancers14071786

**Published:** 2022-03-31

**Authors:** Julia Simões Corrêa Galendi, Sibylle Kautz-Freimuth, Stephanie Stock, Dirk Müller

**Affiliations:** Institute of Health Economics and Clinical Epidemiology, Faculty of Medicine and University Hospital of Cologne, University of Cologne, 50935 Cologne, Germany; sibylle.kautz-freimuth@uk-koeln.de (S.K.-F.); stephanie.stock@uk-koeln.de (S.S.)

**Keywords:** cost-effectiveness, patient-centered care, economic modeling, genetic testing, breast cancer, risk-reducing surgery

## Abstract

**Simple Summary:**

For women who have tested positive for *BRCA* mutations, the decision to make use of preventive surgical options, such as risk-reducing mastectomy (RRM) or risk-reducing bilateral salpingo-oophorectomy (RRSO), depends on the women’s personal preferences and the cultural/social context. Among others, the cost-effectiveness of RRM and RRSO can be affected by the uptake rate of these preventive surgical options. Uptake rates of surgery should be given more attention in the conceptualization of health economic modeling studies for RRM and RRSO. Prospective multicenter studies are recommended to reflect regional and national variations in women’s preferences for preventive surgery.

**Abstract:**

The cost-effectiveness of genetic screen-and-treat strategies for women at increased risk for breast and ovarian cancer often depends on the women’s willingness to make use of risk-reducing mastectomy (RRM) or salpingo-oophorectomy (RRSO). To explore the uptake rates of RRM and RRSO applied in health economic modeling studies and the impact of uptake rates on the incremental cost-effectiveness ratios (ICER), we conducted a scoping literature review. In addition, using our own model, we conducted a value of information (VOI) analysis. Among the 19 models included in the review, the uptake rates of RRM ranged from 6% to 47% (RRSO: 10% to 88%). Fifty-seven percent of the models applied retrospective data obtained from registries, hospital records, or questionnaires. According to the models’ deterministic sensitivity analyses, there is a clear trend that a lower uptake rate increased the ICER and vice versa. Our VOI analysis showed high decision uncertainty associated with the uptake rates. In the future, uptake rates should be given more attention in the conceptualization of health economic modeling studies. Prospective studies are recommended to reflect regional and national variations in women’s preferences for preventive surgery.

## 1. Introduction

In recent decades, genetic testing and counseling have evolved to become an essential part of hereditary breast cancer (BC) and ovarian cancer (OC) prevention. Women who are carriers of germline *BRCA1* and/or *BRCA2* mutations can be offered risk management strategies that can significantly reduce the risk of BC/OC and cancer-related mortality. Risk-reducing mastectomy (RRM) has been shown to decrease the risk of BC and to provide an overall survival benefit for *BRCA1* mutation carriers [1]. Risk-reducing salpingo-oophorectomy (RRSO) decreases the risk of OC and improves cancer-related and overall survival [2], while there are inconsistent results on the impact of RRSO on BC risk in *BRCA* mutation carriers [2,3,4]. When opting for one or both risk-reducing surgeries, a woman has to weigh the benefit of reducing the cancer risk against potential negative consequences of these procedures, such as the loss of fertility, premature menopause, or psychological and physical suffering [5,6]. Hence, some women might opt either for delaying preventive surgery or for intensive surveillance instead.

The reimbursement of risk management strategies for *BRCA* mutation carriers depends on their clinical effectiveness for preventing cancer as well as on their economic consequences. To evaluate the lifelong health economic impact of preventive strategies, models are usually applied with several input parameters. These parameters include data on the course of the disease, such as the cancer incidence, the impact of preventive surgeries in reducing cancer risk, costs, and utilities (i.e., health state preference values).

A recent systematic review of health economic modeling studies concluded that targeted screening followed by risk-reducing strategies might be cost-effective. However, the input parameters applied to these models often differed, notably the uptake rates of surgeries (i.e., the women’s choice for RRM and RRSO) [7]. The uptake of surgery among *BRCA* mutation carriers varies substantially around the world. According to data from an international database from 10 countries, the rate of RRM was highest in the United States (50%) and lowest in Poland (4.5%). The uptake of RRSO was highest in France (83%) and lowest in China (37%) [8]. Thus, to some extent, the heterogeneity of uptake rates could be explained by cultural differences across countries [7]. However, the study designs used for measuring women’s uptake rates or the strength of recommendation for prophylactic surgeries in the clinical management of *BRCA* carriers might also explain the observed differences. In addition, there is some evidence that uptake rates of surgeries are sensitive parameters, resulting in potential uncertainty for the model outcomes [7].

To better illuminate how uptake rates are depicted in health economic modeling studies for preventing BRCA-induced cancer and to investigate the relevance of uptake rates for model results, we conducted a scoping review. The objectives of this review were (i) to systematically assess the sources of input data and assumptions for applying uptake rates of surgery within cost-effectiveness modeling studies and (ii) to assess the degree of uncertainty in the model outcomes that may result from different uptake rates in different settings. In addition, we conducted a value of information (VOI) analysis, based on one previously published model, to exemplify the decision uncertainty that results from uncertainty in the model outcomes [9].

## 2. Materials and Methods

### 2.1. Literature Review

The reporting of this scoping review is in accordance with the preferred reporting items for systematic reviews and the meta-analyses extension for scoping reviews (PRISMA-ScR) checklist [10]. The protocol for this scoping review was not pre-registered. A literature search was conducted in MEDLINE (via PUBMED) on 21 September 2022 and the Centre for Reviews and Dissemination (CRD) database to search for health economic modeling studies that addressed women who were offered RRM and/or RRSO after screening for germline *BRCA* mutations. In addition, we screened the studies included in the most recent systematic reviews published on the topic [7,11,12,13,14]. The search strategy is provided in the Appendix A. Two reviewers screened the titles/abstracts of studies and selected potential studies for full text reading. The study selection and data extraction were carried out independently, and, in case of disagreement, consensus was achieved by discussion.

We included cost-effectiveness modeling studies that (i) targeted women at high clinical or familial risk for carrying *BRCA* mutations or known carriers of *BRCA* mutations and provided genetic testing for inheritable germline mutations including but not limited to *BRCA* mutations, (ii) evaluated risk management strategies based on RRM and/or RRSO, and (iii) presented the model outcomes as incremental cost-effectiveness ratios. Studies were excluded if the reporting of the uptake rates was insufficient, or if the uptake rates were based on an assumption of perfect adherence. There was no language restriction.

From the selected models, we extracted the study characteristics (e.g., strategies used for comparison, model population) and the applied rates of the uptake of surgery (and respective age) in the case of a positive gene test result. In addition, we extracted results from deterministic sensitivity analyses to assess the impact of varying uptake rates of surgery on the incremental cost-effectiveness ratio (ICER). In order to assess if the uptake rates were appropriate for the models’ target population, the cited sources were retraced, from which we extracted data with regard to the study design, setting, number of participants, and time of follow-up.

### 2.2. Value of Information (VOI) Analysis

Based on a model developed and previously published by our institution [9], we conducted a VOI analysis to estimate whether the costs of additional evidence (e.g., conducting a new study) for reducing decision uncertainty associated with model outcomes are worthwhile. The model, on which the VOI analysis was based, assessed the cost-effectiveness of screen-and-treat strategies for German women at risk of hereditary BC and OC versus no testing. The model had a lifelong time horizon and included the health states ‘well’, ‘breast cancer without metastases’, ‘breast cancer with metastases’, ‘ovarian cancer’, ‘death’, and two post (non-metastatic) breast or ovarian cancer states. The perspective of the German statutory health insurance (SHI) was adopted, and input data were predominantly taken from German sources. While the input data concerning uptake rates are reported in Table 1 (i.e., Müller 2018), all input data are reproduced in the Appendix A [9].

The expected value of perfect information (EVPI) is computed as the difference in terms of the net monetary benefit (NMB) between the expected value of a decision made with perfect information and the expected value of the decision based on the current evidence [15]. While the EVPI shows the overall uncertainty, the expected value of partial perfect information (EVPPI) determines which parameters are highly related to decision uncertainty and the potential value of reducing that uncertainty by collecting more data on these specific parameters [16]. 

The 10,000 iterations generated in the probabilistic sensitivity analysis from our model were entered into the Sheffield Accelerated Value of Information (SAVI), which consists of a regression-based method for the EVPI and EVPPI calculations [17]. The value of eliminating parameter uncertainty associated with the uptake rates was quantified in comparison to three sets of other relevant model parameters—utilities (i.e., the quality-adjusted life year values), cancer incidence on *BRCA* mutation carriers, and risk reduction of preventive surgeries. These parameter sets were chosen due to their relevance in deterministic sensitivity analyses [9]. The NMB, which indicates the value of an intervention in monetary terms, was calculated for a hypothetical willingness to pay EUR 10,000.

## 3. Results

After the removal of duplicates, the search yielded 1197 references. After reading titles and abstracts, 31 studies were selected for full text reading. Among these, nineteen health economic modeling studies fulfilled the inclusion criteria [9,18,19,20,21,22,23,24,25,26,27,28,29,30,31,32,33,34,35]. Four studies were excluded due to insufficient reporting of the uptake rates of risk-reducing surgeries [36] because a perfect uptake of surgery was assumed [37,38] or because of an inappropriate presentation of the model result [39]. More information on the excluded studies is provided in the Appendix A. Two studies that were included had not yet been considered in any of the screened systematic reviews [22,31]. Figure 1 shows a flowchart of the study selection process.

The health economic models included covered health systems from different countries, including Norway [19], Australia [20,31,35], Brazil [32,33], the United Kingdom [18,21,25,34], the United States [22,23,28], Canada [24,26,27], Spain [30], and Germany [9]. Table 1 provides an overview of the included models and their uptake rates. 

### 3.1. Strategies Being Compared

A screen-and-treat intervention comprising *BRCA* genetic testing (i.e., full sequencing of *BRCA* genes) followed by RRM and/or RRSO was compared with a no prevention strategy by 10 studies [9,20,21,23,24,25,30,32,33,35]. A reference model developed by NICE compared testing vs. no testing; in this model, a proportion of women received risk-reducing surgery independent of the provision or outcome of testing [18]. In addition, in seven models, risk-reducing surgery was offered to both intervention and controls with differences between the compared strategies: Two models compared two testing strategies, namely full sequencing of *BRCA* genes versus a 7- or 14-gene panel [19,28], while five studies compared testing women based on familial/clinical risk versus different populational criteria [22,26,27,29,34]. While most studies provided immediate surgery for women who had tested positive, five studies modeled a woman’s option to delay surgery [19,22,26,30,33].

### 3.2. Study Population

In eight models, the model population was composed of index patients (i.e., the first person in the family diagnosed with a *BRCA* mutation after a diagnosis of either BC or OC), followed by cascade testing of first- and second-degree healthy relatives [18,19,21,24,25,30,34,35]. Nine models addressed healthy women at increased risk for *BRCA* mutations due to familial risk [18,19,21,24,25,30,34,35]. Whereas some studies limited the population to first- or second-degree relatives of women affected by cancer with *BRCA* mutations [20,27,32,33], others defined the population by an established familial risk (with or without a known mutation in the family) [9,22,23,28,29]. Kwon et al. included only index patients diagnosed with BC at different ages [26].

Women entered the models at different ages, varying from 10 years [35] for siblings and children to 55 years [19] for healthy women and from 40 [26] to 55 years [19] for index patients. In most models, risk-reducing surgery was offered immediately after entering the model, while in five studies, the possibility of delaying surgery was accounted for [19,22,26,30,33].

### 3.3. Uptake Rates Applied to the Models

The uptake rate of RRM applied to the included health economic models ranged from 6% [9] to 47% [34], while those of RRSO varied between 10% [19] and 88% [21]. Figure 2A,B illustrates the variability in the uptake rates of RRM and RRSO for two different age groups, whereas the actual rates are described in Table 1.

Whereas in some studies, the uptake rates were obtained from local centers or smaller departments [9,18,21,25,28,30,34], in others, the information was obtained from multicentric studies or registries reflecting larger regions of a country [19,24,27,32,33,35]. The rates were obtained from a single study [19,23,25,28,30,32,33,35] or from multiple studies [18,24,26,29,34]. Uptake rates without providing a reference or based on unpublished data were found in three studies [20,21]. Most studies considered country-specific evidence, with the exception of four [25,29,32,33]. Considerations with regard to the appropriateness of the selected uptake rates for the models’ target population were missing in all studies.

### 3.4. Sources of Uptake Rates

The sources of uptake rates were published between 2000 [40] and 2014 [41,42,43]. Most studies were based on retrospective data obtained from registries, hospital records, or questionnaires (*n* = 10). In seven models, the rates were based on prospective studies with women followed from 1 to 11 years [44] or based on a systematic review [45]. In these studies, women were recruited from the United States [42,44,46,47,48,49,50], the United Kingdom [42,46,50,51,52,53], Australia [54], Spain [41], Canada [43,45,48,49,55,56], and the Netherlands [40]. Two studies included women from several countries (i.e., Austria, Canada, France, Israel, Italy, Norway, Poland, and the United States) [48,49]. Table 2 details the methodological characteristics of the sources of data regarding uptake rates (as cited in each model study included).

The lowest uptake rate of RRM among healthy *BRCA* mutation carriers was reported by Metcalfe et al. in a retrospective cohort of 177 Norwegian women [48]. In that study, only 5% opted for RRM during the study follow-up [48]. In contrast, 51% of the 257 women retrospectively followed by Meijers-Heijboer et al. opted for RRM as the preferred strategy [40].

With regard to RRSO, the highest uptake was reported by Chai et al. (i.e., 86% of *BRCA1* and 70% of *BRCA2* mutation carriers under 50 years) [42]. In this study, all women were unaffected by cancer. In contrast, the lowest uptake of RRSO (26%) was reported by a retrospective single-center study that addressed women with previous BC to prevent a recurrent or contralateral BC [51].

### 3.5. Impact of Varying Uptake Rates in Sensitivity Analyses

Most studies provided information about the impact of varying the uptake of surgery in a deterministic sensitivity analysis, except for three [25,27,32]. In all but one of these studies [21], higher uptake rates improved the incremental cost-effectiveness ratio (ICER). Table 3 summarizes the results of the deterministic sensitivity analysis reported by the models. 

In seven studies, the impact of varying uptake rates on the cost-effectiveness was remarkable [9,21,23,24,26,33,35]. For instance, varying uptake rates changed the cost-effectiveness ratio from 20% to 40% [21] to more than 70% [9]. A common aspect of these studies is that the strategies being compared comprise a screen-and-treat intervention versus a no-testing strategy (i.e., no surgery in the comparator arm). In contrast, in six of the modeling studies, the authors considered the impact of varying the uptake rates to be slight (≤10%) or negligible [18,19,20,22,29,34]. In all of these studies, a risk-reducing surgery was offered to both the intervention and the comparator arm [18,19,22,29,34].

### 3.6. VOI Analysis

The overall EVPI per person is estimated at EUR 1680, which is the value of acquiring perfect information (i.e., eliminating all uncertainty) about all parameters applied to the model (detailed in Appendix A). The EVPPI per person for the predefined parameter sets is shown in Figure 3. The EVPPI value indicates to what extent more information on these sets of parameters would reduce the decision uncertainty (i.e., the chance that the decision-maker incorrectly opts for the strategy with lower payoffs, which, in our model, was the no-testing strategy). The maximum return in terms of the net monetary benefit from removing uncertainty around the uptake rates was EUR 239 (standard error (SE): EUR 24), corresponding to 14% of the total EVPI. The second set of parameters with the highest EVPPI was cancer incidence in the BRCA mutation carriers (EUR 207, SE: 25), followed by the risk reduction of preventive surgeries (EUR 188, SE: 25), and the lowest were the utilities (EUR 154, SE: 27). 

## 4. Discussion

According to this comparison, the uptake rates of risk-reducing surgeries applied in cost-effectiveness models are sensitive parameters. In the models’ deterministic sensitivity analyses, there was a clear trend that a lower uptake rate increased the ICER and vice versa. Considering the vast potential of both RRM and RRSO for reducing the risk of cancer and cancer-related mortality, this is a little surprising. However, in one analysis, the authors reported a slightly higher cost-effectiveness ratio compared to the base case when the uptake of RRM was increased in a sensitivity analysis. The authors explained this counterintuitive effect with high costs for preventive treatment, which were not offset by survival gains (because of the high survival rates in women who do not undergo RRM) [21].

While deterministic analyses demonstrate the model’s sensitivity to a single input parameter, the VOI analysis evaluates the uncertainty of multiple parameters simultaneously. By sampling each parameter several times from a given range at each iteration, a more reliable estimation of the uncertainty can be provided, especially in models with parameters that correlate to each other [15]. The VOI analyses indicate the potential NMB forgone by making the decision between two treatment alternatives with current (i.e., uncertain) parameters, in comparison to making the decision with perfect information. As a decision rule for VOI analyses, the cost of future studies to gather more information about uncertain model parameters should not exceed the NMB elicited in the VOI analysis [15].

The high EVPPI of uptake rates indicates that gathering more information about the uptake rates would have a slightly higher impact on reducing decision uncertainty than additional information about other parameter sets (i.e., cancer incidence on *BRCA* mutation carriers, risk reduction of preventive surgeries, or utilities). Although the VOI calculation reflects the uncertainty in the German model, this finding is likely to be replicated in similar models. The VOI analysis can be easily replicated using regression-based methods based on the iterations generated in the probabilistic sensitivity analysis [17].

The uptake rates applied to modeling studies varied substantially. To reflect the attitudes and preferences of the different target populations, different sources of input data have been chosen for models. This variability might be explained by several factors, such as (i) cultural differences, (ii) individual-related factors, (iii) age-dependent factors, and (iv) an improved acceptance of preventive surgeries over time.

(i) Cultural differences (e.g., perception of health and disease, femininity, autonomy) and the risk of financial and social discrimination might influence the preference for genetic testing and risk-reducing surgery [40]. In a previous systematic review, it was suggested that cultural differences between countries could explain the variability in uptake rates to a large degree [7]. However, according to our results, most models used country-specific data, and there was even considerable variability within countries. For instance, among studies conducted in the UK, the rates of RRM varied from 0.21 [46] to 0.43 [42], while those for RRSO varied from 0.26 [58] to 0.86 [42]. Similarly, studies conducted in the United States showed that the uptake rates of RRM varied from 0.36 [45] to 0.42 [42,50], and those for RRSO from 0.33 to 0.71 [48].

(ii) Individual-related factors are also prone to affecting the preferences of women towards RR surgeries. Individual factors that increase the uptake of RRSO include a personal history of BC [49,50,52], parity [44,46,52], and a woman’s postmenopausal status [52], while the uptake of RRM tends to be higher among both parous women and those who have a first-degree relative with BC [44,46]. In addition, women who had a family history of OC were more likely to undergo any surgical option [50].

Moreover, many of these individual factors are (iii) age-dependent. While women who have tested positive for *BRCA1/2* should consider an RRSO by the age of 35 or right after completion of childbearing [59,60], a prospective study shows that the usage of RRM and RRSO occurs later than recommended [42]. The proportion of women that opt for a risk-reducing surgery increases after age 40 probably because fertility is no longer a concern, and the cumulative risk of cancer is more paramount [42]. Accordingly, the uptake rates applied by the models were, in general, lower for women younger than 35 years, with the lowest uptake rate applied for RRSO (10%) in women younger than 35 [19]. 

Finally, recent evidence indicates (iv) an improved acceptance of preventive surgeries over time. A reason for this trend could be the improvement in genetic counseling protocols and the cross-center knowledge transfer [61]. Increased uptake rates of risk-reducing surgeries over time due to improved adherence have been observed for the uptake of RRM in women with *BRCA* mutations, while the uptake of RRSO remained stable [58]. Nevertheless, the trend for RRM was not confirmed by the modeling studies included in our review.

As a limitation of this literature review, it should be acknowledged that there was no protocol registration, and a critical appraisal within sources of evidence was not conducted. Furthermore, because sources of uptake rates were identified only if used for a cost-effectiveness model, it is not possible to draw firm conclusions on temporal, regional, or cultural trends or individual factors. To evaluate these relationships more precisely, a comprehensive literature review of observational studies has to be performed. However, our review could demonstrate how sensitive models were when depicting the complexity inherent to the uptake rates.

The usage of outdated sources of evidence for decision-making carries substantial uncertainty regarding the payer’s outcomes. The improved counseling for *BRCA* mutation carriers in recent years might have gradually reduced women’s reluctance in opting for risk-reducing surgery, resulting in higher uptake rates. Hence, country-specific, prospective, multi-center studies including post-testing counseling with respect to age and subgroups should be performed to reflect the current status of women’s preferences for or against surgical prevention. However, as long as updated evidence is not available, modelers—at least in some countries—have to rely on data obtained from retrospective surveys, cross-sectional studies, or medical records without accounting for follow-up. In this case, assessing uncertainty associated with the uptake rates applied to the models is of utmost importance to provide the decision-maker with a realistic assessment of the economic consequences when adopting a screen-and-treat strategy for women with *BRCA* mutations.

## 5. Conclusions

The uptake rates of risk-reducing surgeries applied to modeling studies assessing the cost-effectiveness of screen-and-treat strategies vary considerably. Uptake rates of surgery are associated with high uncertainty, especially in modeling studies comparing a screen-and-treat intervention versus a no-testing strategy. Country-specific and prospective studies including non-directive counseling should be performed to reflect women’s preferences for or against surgical prevention and would provide a stronger evidence base for economic modeling studies.

## Figures and Tables

**Figure 1 cancers-14-01786-f001:**
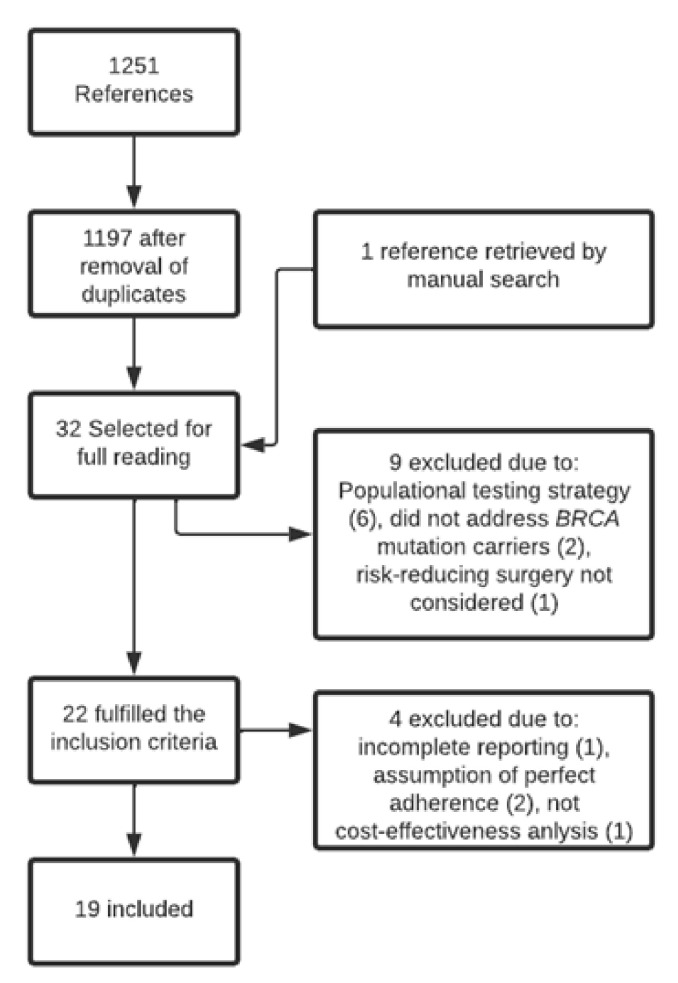
Study selection process.

**Figure 2 cancers-14-01786-f002:**
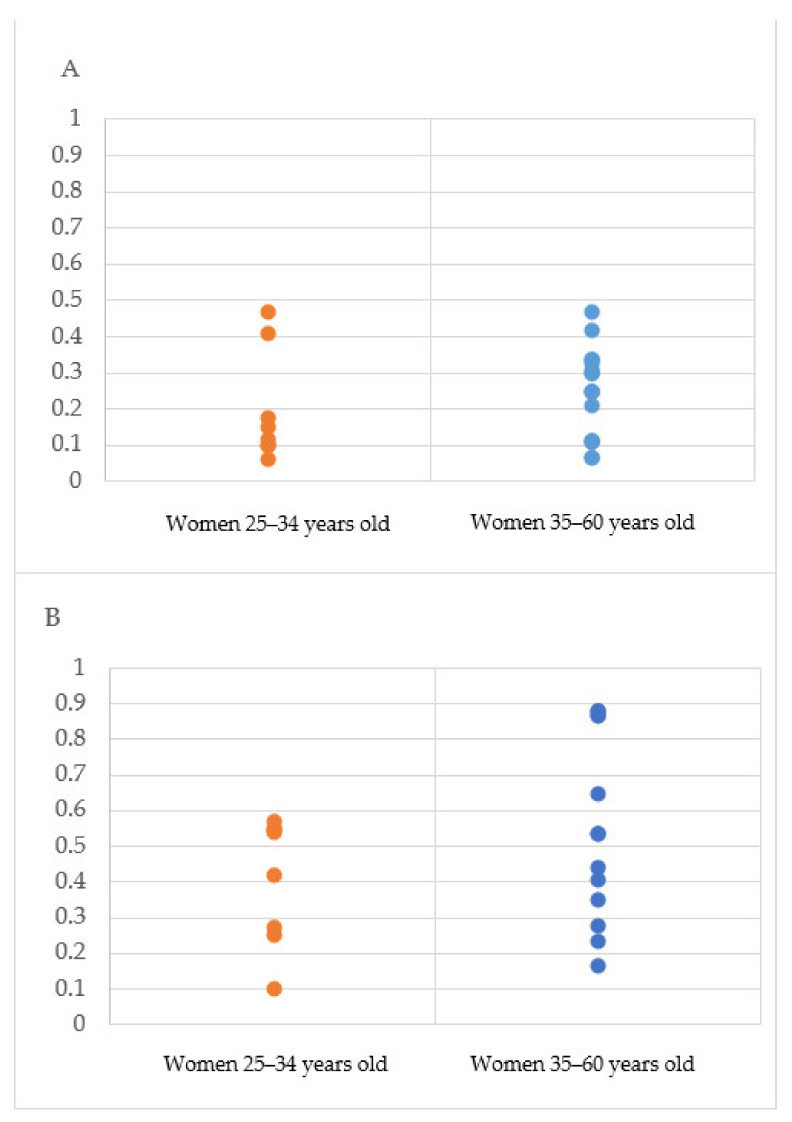
Variability of the uptake rates applied to the models in two age groups. (**A**) Uptake rates of risk-reducing mastectomy; (**B**) uptake rates of risk-reducing salpingo-oophorectomy.

**Figure 3 cancers-14-01786-f003:**
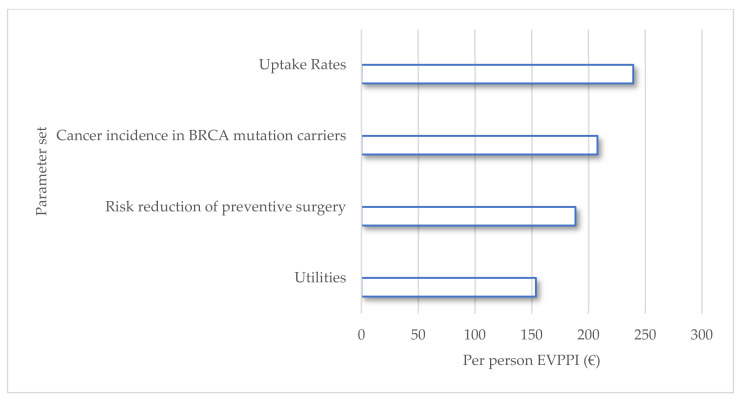
Expected value of partial perfect information (EVPPI) for parameter sets.

**Table 1 cancers-14-01786-t001:** Characteristics of modeling studies included.

Author/Year	Country	Strategies Being Compared	Model Population	RRM Uptake Rate (Age, in Years)	RRSO Uptake Rate (Age, in Years)	Combined RRM and RRSO
Müller 2018 † [9]	Germany	Testing (sequencing of *BRCA**1/2*) vs. no testing	Women at risk for hereditary BC or OC due to family history, entering the model at age 35	0.06 (35)	0.42 (35)	0.45 (35)
Simões Correa Galendi 2020 [33]	Brazil	Testing (sequencing of *BRCA1/2*) vs. no testing	First-degree relatives of index patients (BC or OC) with *BRCA**1/2* mutations, entering the model at age 30	0.10 (30–34) 0.11 (35–39) 0.07 (40)	0.27 (30–34) 0.23 (35–39) 0.17 (40)	Not considered
Petelin 2020 [31]	Australia	Risk management strategy (including risk-reducing surgeries) vs. population-based breast screening program	*BRCA**1/2* mutation carriers entering the model at age 20	0.31(39)	0.41 (45)	
Manchanda 2020 [29]	United Kingdom/USA/Netherlands China/Brazil India	Testing (sequencing of *BRCA**1/2*) all general population women ≥ 30 years vs. clinical criteria/FH-based testing	Women at risk for having mutations based on clinical and FH, entering the model at age 30	0.47	0.55	Not considered
Hurry 2020 [24]	Canada	Testing (sequencing of *BRCA**1/2*) vs. no testing	Index patients aged 50; first- and second-degree relatives (daughters entered the model at age 20, sisters at 50)	0.21 (44)	0.44 (54)	Not considered
Guzauskas 2020 [22]	United States	Population-based testing (sequencing of *BRCA**1/2*) vs. testing based on FH or clinical risk	Women at risk for having mutations based on clinical and FH, entering the model at age 30 or 45	0.10 (30–34) 0.11 (35–39) 0.07 (40)	0.27 (30–34) 0.23 (35–39) 0.17 (40)	Not considered
Sun 2019 [34]	United Stated and United Kingdom	Testing (sequencing of *BRCA**1/2*) for all women with BC vs. based on FH or clinical risk	Index patients (BC); first-degree relatives of index patients with *BRCA*1/2 mutations, entering the model at different ages ¥	0.47(30)	0.55 (30)	Not considered
Moya-Alarcón 2019 [30]	Spain	Testing (sequencing of *BRCA**1/2*) vs. no testing	Index patients at age 51 (OC); first- and second-degree relatives (daughters, nephews and nieces entered the model at age 23)	0.25 (45–55)	0.65 (45–55)	Not considered
Kwon 2019 [27]	Canada	Testing followed by RRSO (Sequencing of *BRCA1/2*) vs. no testing vs. RRSO for all (without testing)	First-degree relatives of index patients (OC), entering the model at age 40	Not considered	0.54 (40–50)	0.33 (40–50)
Kemp 2019 [25]	United Kingdom	Testing (sequencing of *BRCA**1/2*) vs. no testing	Index patients aged 50 years (BC); first- and second-degree relatives (daughters entered the model at age 20, sisters at 50)	*BRCA1* 0.34 (40) *BRCA2* 0.25 (40)	*BRCA1* 0.88 (40) *BRCA2* 0.87 (40)	Not considered
Asphaug 2019 [19]	Norway	Full sequencing of *BRCA1/2* vs. seven-gene panel vs. 14-gene panel	Index patients aged 55 years (BC); first-degree relatives (daughters entered the model at age 25 and sisters at 55)	0.12 (25–34) 0.11 (35–60)	0.10 (25–34) 0.28 (35–39) 0.35 (40–60)	Not considered
Tuffaha 2018 [35]	Australia	Testing (sequencing of *BRCA1/2*) vs. no testing	Index patients at age 40 (BC) with 10% probability for *BRCA**1/2* mutations; first- and second-degree relatives (children entered the model at age 10, siblings at age 40)	0.3 (40)	0.54 (40)	0.16 (40)
Ramos 2018 [32]	Brazil	Testing (sequencing of *BRCA**1/2*) vs. no testing	First-degree female relatives of index patients (OC) with *BRCA**1/2* mutations, entering the model at age 30	0.18 (30)	0.57 (30)	Not considered
Li 2017 [28]	United States	Full sequencing of *BRCA**1/2* vs. five-gene panel	Women at risk for hereditary BC or OC due to family history or other hereditary syndromes, entering the model at age 40 or 50	0.42 (50)	Not considered	Not considered
Eccleston 2017 [21]	United Kingdom	Testing (sequencing of *BRCA**1/2*) vs. no testing	Index patients age 50 years (OC) First- and second-degree relatives (daughters entered the model at age 20, sisters at 50)	*BRCA**1* 0.34 (40) *BRCA*2 0.25 (40)	*BRCA**1* 0.88 (40) *BRCA*2 0.87 (40)	Not considered
NICE 2013 [18]	United Kingdom	Testing (sequencing of *BRCA**1/2*) vs. no testing	First-degree female relatives of index patients (BC or OC) with *BRCA**1/2* mutations, entering the model at different ages 20–70	0.42 (30)	0.54 (35)	0.15
Kwon 2010 [26]	Canada	Testing (different criteria for sequencing of *BRCA**1/2*) vs. no testing	Subgroups of women with BC before age 40 or 50, regardless of ethnicity of family history	0.20 (50–55) §	0.55 (50–55)	Not considered
Holland 2009 [23]	United States	Testing (sequencing of *BRCA**1/2*) vs. no testing	Women with 10% pre-test probability of having a mutation, ‡ who entered the model at age 35	0.15 (35)	0.25 (35)	Not considered
Breheny 2005 [20]	Australia	Testing (sequencing of *BRCA**1/2*) vs. no testing	First-degree relatives of individuals with *BRCA**1/2* mutations, entering the model at age 25	0.30 (38)	-	Not considered

Abbreviations: BC: breast cancer, OC: ovarian cancer, RRM: mastectomy, RRSO: salpingo-oophorectomy, FH: family history. † Model used for value of information analysis; ‡ implies some familial history, but not necessarily a known mutation in the family; § in this population, RRM referred to contralateral mastectomy, assuming unilateral mastectomy as first-line BC treatment; ¥ individual simulation with clinical trial data.

**Table 2 cancers-14-01786-t002:** Sources of uptake rates cited by the included health economic models.

Author/Year	Source of Uptake Rate (Year)	Study Design	Country	Number of Participants	Follow-Up
Müller 2018 † [9]	Unpublished	Cross-sectional (single-center, hypothetical responses of women in a counseling situation)	Germany	136 women at different ages following individual genetic counseling	-
Simões Correa Galendi 2020 [33]	Chai (2014) [42]	Prospective, multi-center (post-testing counseling)	United States, United Kingdom	1499 healthy women with inherited *BRCA**1/2* mutations	At least 0.5 years
Petelin 2020 [31]	Petelin (2019) [57]	Prospective and retrospective collected clinical data from a single specialized cancer center	Australia	983 women with *BRCA**1/2* mutations (302 had BC at diagnosis)	6.5 years
Manchanda 2020 [29]	Evans (2009) [58]	Matched controls (regional cancer registries)	United Kingdom	221 healthy women with known *BRCA**1/2* mutations	7 years
Hurry 2020 [24]	RRM: Metcalfe (2007) [56] RRSO: McAlpine (2014) [43]	Retrospective (databases of mutation carriers Hospital discharges (RRSO)	Canada	RRM: 342 women with *BRCA* mutations, healthy and previous BC RRSO: 2119 who underwent hysterectomy (with or without BSO) or BSO or sterilization	RRM: 4 years RRSO:
Guzauskas 2020 [22]	Chai (2014) [42]	Prospective, multi-center (post-testing counseling)	United States, United Kingdom	1499 healthy women with inherited *BRCA**1/2* mutations	At least 6 months
Sun 2019 [34]	RRM: Evans (2009) [58] RRSO: Manchanda (2012) [52]	Matched controls (regional cancer registries) Prospective observational cohort	United Kingdom	RRM: 105 women with *BRCA* mutations (healthy and BC) RRSO: 1133 women at high risk, less than 50% had *BRCA* mutations	7 years 6 years
Moya-Alarcón 2019 [30]	Esteban (2015) [41]	Retrospective (hospital data)	Spain	969 women from 682 families	
Kwon 2019 [27]	Metcalfe (2008) [48]	Retrospective (multicenter study, questionnaire after receiving genetic test)	United States	RRSO: 703 women, healthy and with previous BC with *BRCA* mutations	3.9 years
Kemp 2019 [25]	-	Retrospective (unpublished single hospital data)	United Kingdom.	858 women with *BRCA* mutations (unclear if previous cancer diagnosis)	-
Asphaug 2019 [19]	Metcalfe (2008) [48]	Retrospective (multi-center, questionnaire after receiving genetic test)	Austria, Canada, France, Israel, Italy, Norway, Poland, United States	RRM: 1290 RRSO: 177 women, healthy and with previous BC with *BRCA* mutations	3.9 years
Tuffaha 2018 [35]	Collins (2013) [54]	Prospective (multicenter, interviewer-administered questionnaire, surgery confirmed from pathology and medical records)	Australia	325 healthy women with inherited *BRCA* mutations	3 years
Ramos 2018 [32]	Metcalfe (2008) [48]	Retrospective (multicenter, questionnaire after receiving genetic test)	Various, Canada	RRM: 766/RRSO: 1383 women, healthy and with previous BC, with *BRCA* mutations	3.9 years
Li 2017 [28]	Singh (2013) [44]	Retrospective (registry data)	United States	136 women with inherited *BRCA* mutations without previous cancer diagnosis	1–11 years
Eccleston 2017 [21]	-	Retrospective (unpublished single hospital data)	United Kingdom	858 women with *BRCA* mutations (unclear if previous cancer diagnosis)	-
NICE 2013 [18]	RRM: Evans (2009) [58] RRSO: Sidon (2012) [53] RRSO/RRM: Uyei (2006) [50]	Matched controls (regional cancer registries) Retrospective (regional cancer registries) Retrospective (medical records)	United Kingdom	RRM: 105 RRSO: 314 RRM/RRSO: 554 All women with *BRCA* mutations, healthy or with BC	7 years 5 years 6 years
Kwon 2010 [26]	RRM: Metcalfe (2004) [47]	Retrospective (medical records)	United States, the Netherlands	Metcalfe (2004): 390 women with early-stage BC, who are known carriers or are likely to carry *BRCA1/2* mutations and were treated with unilateral mastectomy Other studies: healthy women with *BRCA* mutations and diagnosis of BC	9 years
	RRSO: Friebel (2007) [46]	Prospective (questionnaire, medical records)			
	Meijers-Heijboer (2000) [40]	Prospective (single-center, hospital data)			
	Metcalfe (2008) [48]	Retrospective (multicenter, questionnaire after receiving genetic test)			
	Metcalfe (2008) [49]	Prospective (multicenter, questionnaires)			
Holland 2009 [23]	Weinberg (2004) [45]	Meta-analysis (five studies for uptake of BC, six studies for uptake of OC)	Various	354 healthy, pre-symptomatic women who knew their mutation status and who had no prior history of BC or OC	
Breheny 2005 [20]	-	Provided abbreviation not identifiable	-	-	

Abbreviations: BC: breast cancer, OC: ovarian cancer, RRM: risk-reducing mastectomy, RRSO: risk-reducing salpingo-oophorectomy. † Model used for value of information analysis.

**Table 3 cancers-14-01786-t003:** Results of deterministic sensitivity analysis reported by the included models.

Author/Year	Strategies Being Compared	ICER	Deterministic Sensitivity Analysis (Impact on the ICER by Varying the Uptake Rates)
Müller 2018 † [9]	Testing vs. no testing	EUR 17,027/QALY	5% lower uptake of RRSO and RRSO combined with RRM increased ICER by 70%.
Simões Correa Galendi 2020 [33]	Testing vs. no testing	BRL 24,264/QALY (USD 11,726/QALY)	10% lower uptake rates of all risk-reducing surgeries increased the ICER by 10%; 20% lower uptake rates of all RR surgeries increased the ICER by 30%.
Petelin 2020 [31]	Risk management strategy vs. population-based breast screening program	AUD 32,359/QALY (*BRCA1*) AUD 48,263/QALY (*BRCA2*)	At a 75% reduced uptake of RRSO, the ICER increased by 25% and 15% for *BRCA1* and *BRCA2* mutation carriers, respectively. At a 75% reduced uptake of RRM, the ICER decreased by 1% and 17% for *BRCA1* and *BRCA2* mutation carriers, respectively.
Manchanda 2020 [29]	Populational testing vs. clinical criteria/FH-based testing	UK: USD 21,191/QALY USA: USD 16,552/QALY NL: USD 25,215/QALY China: USD 23,485/QALY Brazil: USD 20,995/QALY India: USD 32,217/QALY	Half the uptake rate for RRM or RRSO increased the ICER by about 5%.
Hurry 2020 [24]	Testing vs. no testing	CAD 14,294/QALY (USD 10,555/QALY)	50% increase in RRS uptake rates (RRSO 0.66 and RRM 0.32), and mean age of RRSO 50 years reduced the ICER 85%.
Guzauskas 2020 [22]	Population-based testing vs. testing based on FH or clinical risk	USD 87,700/QALY	Considering an uptake rate of RRSO or RRM of 50% lower (or 50% higher) increased (or reduced) the ICER by 10%.
Sun 2019 [34]	Testing for all women with BC vs. based on FH or clinical risk	UK: GBP 10,464/QALY USA: USD 65,661/QALY	10% higher uptake of RRSO reduced the ICER by 10%, and 10% lower uptake increase the ICER by 10% (for the UK payer perspective); 10% higher uptake of RRSO increased the ICER by 5%, 10% lower uptake decreased the ICER by 40% (for the US payer perspective).
Moya-Alarcón 2019 [30]	Testing vs. no testing	EUR 31,621/QALY	Considering an uptake rate of RRSO or RRM 25% lower (or 25% higher) increased (or reduced) the ICER by 5%.
Kwon 2019 [27]	Testing followed by RRSO vs. no testing	USD 7888 per QALY	Not reported
Kemp 2019 [25]	Testing vs. no testing	USD 1330/QALY	Not reported
Asphaug 2019 [19]	Full sequencing of *BRCA1/2* vs. seven-gene panel vs. 14-gene panel	USD 53,310/QALY	Considered negligible by the author.
Tuffaha 2018 [35]	Testing vs. no testing	AUD 18,900	Significant Reducing the uptake rates by 10%, the ICER increased 40–50%.
Ramos 2018 [32]	Testing vs. no testing	BRL 908/case of cancer avoided	Not reported
Li 2017 [28]	Full sequencing of *BRCA**1/2* vs. five-gene panel	USD 69,920/QALY	Considering an uptake rate of RRM 50% lower (or 50% higher) increased the ICER by 50% (or reduced the ICER by 40%).
Eccleston 2017 [21]	Testing vs. no testing	GBP 4339/QALY	Considering an uptake rate of RRSO 75% lower increased the ICER by 40%. Considering an uptake rate of RRM 50% higher decreased the ICER by 23%.
NICE 2013 [18]	Testing vs. no testing	GBP 18,114/QALY §	Considered negligible by the author.
Kwon 2010 [26]	Testing vs. no testing	USD 9084/QALY	The ICER increased about 30% when applying a realistic scenario (40% choose no procedure) over an ideal scenario (100% uptake).
Holland 2009 [23]	Testing vs. no testing	USD 9000/QALY	The ICER decreased as the rate of RRM increased and dominated above an 80% RRM rate. Higher rates (until 60%) of RRSO also decreased the ICER, and higher than 60%, the incremental benefits decreased faster than the incremental costs, increasing the ICER.
Breheny 2005 [20]	Testing (sequencing of *BRCA**1/2*) vs. no testing	USD 477/cancer-free year gained (*BRCA1*) USD 2150/cancer-free year gained (*BRCA2*)	Varying the uptake rate of RRM from 0% to 50%, the latter reduced the ICER by 10%.

Abbreviations: ICER: incremental cost-effectiveness ratio, RRM: risk-reducing mastectomy, RRSO: risk-reducing salpingo-oophorectomy. † Model used for value of information analysis. § women aged 40–49 at 10% pre-test probability.

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
