# Peer review of "Uptake Rates of Risk-Reducing Surgeries for Women at Increased Risk of Hereditary Breast and Ovarian Cancer Applied to Cost-Effectiveness Analyses: A Scoping Systematic Review"

_cancers, 2022, doi:10.3390/cancers14071786_

Round 1

Reviewer 1 Report

COMMENTS TO AUTHORS:

This paper explores the application of uptake rates of preventive surgery of women at increased risk for BC and OC due to BRCA mutations in health economic modeling studies. I do have some comments as listed below in the order noted.

Comment 1:

The quality of the data set is very important, please provide the Preferred Reporting Items for Systematic Reviews and Meta analyses (PRISMA) reporting guideline.

Comment 2:

Please provide a new table to conduct the “Comparison of Studies Reporting Final Outcomes.”

Author Response

Thank you for the careful review. We improved the literature review by applying the preferred reporting items for systematic reviews extension for Scoping Reviews. To facilitate the peer review process, we attached the PRISMA-ScR checklist to the supplementary files. From the PRISMA-ScR checklist, two items were not considered, namely the beforehand registration of the review protocol and the critical appraisal of the included studies (the latter is an optional step for a scoping review). The items not addresed were acknowledged in the study limitations (page 13 lines 296-298)

We included a new column in table 3 to report the final outcomes of studies.

Reviewer 2 Report

It was not clear from the title or the abstract that this is mainly a review of uptake rates in cost-effectiveness studies. Please revise the title and abstract to clearly state that this is a review study.

The aims at the end of the intro need to be revised to make it clear this study mainly conducted a systematic review of health economics models.

The necessity of this review is not clearly explained. Variation of the rates used in modeling studies is expected. If all studies used the same rate, is it a good thing? If the authors only want to argue that getting the information about update rate is important, it seems the VOI analysis would be sufficient for that. Why is this review necessary? Did the VOI analysis use any information from the models reviewed?

No details about how the VOI analysis was provided in this paper. Instead, the authors referred the readers to a previous paper that used the model. Only summary results were presented here. However, this analysis is stated as a second aim of this study. If it has already been done in a previous study, then it should not be included as a second aim for this study because it was not done in this paper. If only the same model was used, the authors should still provide more details about the model, how the model was used to arrive at the EPVI or EPPVI value, and what assumptions were used etc. Without them, it is hard to determine if the EPVI or EPPVI values could be applied to other countries.

Please provide justification for why only 39 full text articles were chosen for review from 1197 references. Why were the rest of references not used?

Reference numbers are used in the text but not tables. Suggest adding reference numbers to Tables so that it would be easier for readers to reference the table content while reading the text. Otherwise, readers have to go to the references first to find the articles and then go back to the tables to find relevant information of these articles.

In Table 3, the models did not compare to the same control option. Therefore, the interpretations of the ICERs are different. It would be helpful to include a column showing what comparison was made in an iCER and then in the next column shows the impact of uptake rate on that ICER.

To “what” extend not to “which” extend.

Author Response

Dear reviewer, thank you for the careful review.

All comments are copied here and answered in red. We stay at your disposal for further clarifications.

It was not clear from the title or the abstract that this is mainly a review of uptake rates in cost-effectiveness studies. Please revise the title and abstract to clearly state that this is a review study. The aims at the end of the intro need to be revised to make it clear this study mainly conducted a systematic review of health economics models.

We revised title, abstract (lines 19 and 21) and introduction (lines 64 and 69-71). We also applied the PRISMA-ScR for reporting scoping reviews, as suggested by reviewer 1 (supplementary file attached for the peer review process).

The necessity of this review is not clearly explained. Variation of the rates used in modeling studies is expected. If all studies used the same rate, is it a good thing? If the authors only want to argue that getting the information about update rate is important, it seems the VOI analysis would be sufficient for that. Why is this review necessary? Did the VOI analysis use any information from the models reviewed?

The reviewer is right. We revised the introduction and now more elucidated the rationale of this review (page 2, line 61-73).

No details about how the VOI analysis was provided in this paper. Instead, the authors referred the readers to a previous paper that used the model. Only summary results were presented here. However, this analysis is stated as a second aim of this study. If it has already been done in a previous study, then it should not be included as a second aim for this study because it was not done in this paper. If only the same model was used, the authors should still provide more details about the model, how the model was used to arrive at the EPVI or EPPVI value, and what assumptions were used etc. Without them, it is hard to determine if the EPVI or EPPVI values could be applied to other countries.

The VOI analysis is an additional calculation based on the 10000 iterations generated during the probabilistic sensitivity analysis from the model previously published. Although the model structure is described in detail elsewhere, the input data and assumptions concerning uptake rates are reported within the scoping review. We clarified this in page 3, methods, lines 102-106 and line 114. Alterations were made to the tables to highlight the study that was used for the VOI. The EVPI and EVPPI reflect the uncertainty on this specific model. (Amended on page 13, line 259-260)  

Please provide justification for why only 39 full text articles were chosen for review from 1197 references. Why were the rest of references not used?

Title and abstracts of the 1197 references were screened by reading title and abstract to select the studies that would potentially fulfil our inclusion criteria (added in page 3, line 123-124). The vast majority of the excluded studies were not health economic evaluations, due to the sensibility of our search strategy, which is provided in the supplementary apendix.

Reference numbers are used in the text but not tables. Suggest adding reference numbers to Tables so that it would be easier for readers to reference the table content while reading the text. Otherwise, readers have to go to the references first to find the articles and then go back to the tables to find relevant information of these articles.

Reference numbers added to all tables.

In Table 3, the models did not compare to the same control option. Therefore, the interpretations of the ICERs are different. It would be helpful to include a column showing what comparison was made in an iCER and then in the next column shows the impact of uptake rate on that ICER.

Column added to table 3.

To “what” extend not to “which” extend.

Corrected in page 12, line 226.

Reviewer 3 Report

This paper deals with a literature review on the health economic modeling studies regarding women who were offered RRM and/or RRSO after screening for germline BRCA mutations and the application of the AA model VOI (Value of information) analysis. They concluded that the uptake rates of surgery are associated with high uncertainty in modeling studies considering studies on screen-and-treat interventions versus no strategy but also country specific questions regarding counseling and women preferences. For this reason  health economic modeling studies should pay much more attention to the uptake rates of RRM and RRSO and prospective studies are needed to reflect the regional and national variations in women's preferences.

In this contest I suggest to include also Italian studies as that of Carbonara N et al. J Pers Med 2021 Aug 27;11(9):847.  doi: 10.3390/jpm11090847

Author Response

Thank you for reviewing our manuscript. The study from Carbonara et al presents a cost decision-making model that compares the healthcare costs for diverse treatment strategies for BRCA-mutated women with breast cancer. However, this model is not a cost-effectiveness analysis, it calculates the cancer treatment costs that could potentially be prevented by choosing the treatment strategy with the lowest total cost. Hence, the model outcomes is not an incremental cost-effectiveness ratio, what would make the comparison with other studies difficult. The inclusion criteria was clarified in page 2 line 89 and the exclusion of this study justified in page 3 line 127-128.

Round 2

Reviewer 2 Report

  1. Line 71. What is a difference between an “exemplary” VOI and just a regular VOI?
  2. What does it mean by the “appropriateness” of the uptake rates? Shouldn’t the uptake be 100% since the population for this study are at a high risk?
  3. Line 101, please add reference.
  4. Line 25, please add the explanation why only 31 were included for full text reading out of 1197. It is still not clear.
  5. Although the reference of the previous model was provided, please still add a brief summary of the model structure and discuss any assumptions in the model that is unique to German and may not apply to other countries. Further, please discuss how other countries may modify this model to generate their own VOI estimates.
  6. Please discuss the findings from the VOI. They are presented but no discussion of the meaning of these estimates. Based on the estimates, should a study be conducted to get perfect information on all parameters, and on each parameter?
  7. Figure 3, since details of the VOI model were not provided in this paper, please include the EVPPI estimates for all parameters in the model in Figure 3 so that the readers could better understand “the value of acquiring perfect information about all parameters applied to the model (i.e., eliminating all uncertainty).

Author Response

Dear Reviewer, thank you for reviewing our manuscript. To facilitate the review process, we copied every comment here and answer to them in red. Changes in the manuscript are indicated here and highlighted with track changes in the manuscript. We stay at your disposal for further clarifications. 

--

1. Line 71. What is a difference between an “exemplary” VOI and just a regular VOI?

      There is no difference except that for this study we used the VOI only for a small number of variables to demonstrate the relatively high uncertainty of the uptake rates. Reformulated on lines 73-74.

2. What does it mean by the “appropriateness” of the uptake rates? Shouldn’t the uptake be 100% since the population for this study are at a high risk?

‘Appropriateness’ was reformulated in line 100-101 and line 196 -197.

Because the uptake rates depend also on the women’s’ preferences, a rate of 100% is difficult to achieve. Amended on line 45-46 (introduction)

3. Line 101, please add reference.

Reference added (line 104)

4. Line 25, please add the explanation why only 31 were included for full text reading out of 1197. It is still not clear.

Established methods for searching and selecting relevant articles were applied (PRISMA-ScR). After reading title and abstracts of the 1197 we identified the 31 that would potentially fit our inclusion criteria. We asserted by reading title and abstract that the remaining references did not fit the inclusion criteria of our review. To our knowledge, there is no rule for a minimum of relevant articles given a number of around 1200 hits from the search term.

5. Although the reference of the previous model was provided, please still add a brief summary of the model structure and discuss any assumptions in the model that is unique to German and may not apply to other countries..

More information on the model was provided in lines 110-116.

Further, please discuss how other countries may modify this model to generate their own VOI estimates

Added in lines 279-280

6. Please discuss the findings from the VOI. They are presented but no discussion of the meaning of these estimates. Based on the estimates, should a study be conducted to get perfect information on all parameters, and on each parameter?

Added in lines 268-272

7. Figure 3, since details of the VOI model were not provided in this paper, please include the EVPPI estimates for all parameters in the model in Figure 3 so that the readers could better understand “the value of acquiring perfect information about all parameters applied to the model (i.e., eliminating all uncertainty).

The reviewer is right. We added a full description of all model input parameters to the supplementary file S2 (parameter sets used in the VOI analysis) and S3 (Parameter sets not included in the VOI analysis).

Round 3

Reviewer 2 Report

Thank you for responding to my comments. Please see below a few minor suggestions.

1) The inclusion and exclusion for selecting literature for review were not explicitly stated. It only said the study included these types of articles but it is not clear if these were the initial inclusion and exclusion criteria applied. 

2) Please make it clear on line 242 why the other parameters from 2018 study were used. Although it is implied from the study population, it still would be helpful to point that out.  

Author Response

Dear reviewer, thnk you for reviewing our manuscript. 

With regard to the two minor comments: 

1) Inclusion and exclusion criteria are detailed in lines 88-94. Although we have not published a protocol, there were no changes to the inclusion and exclusion criteria. The lack of a published protocol is also mentioned in the discussion. 

2) Added in lines 129-130 in the methods section.

We stay at your disposal for further clarifications.